# Nutrition Knowledge and Food Literacy Among Persons with Multiple Sclerosis—Development and Validation of Patient-Reported Outcome Measures

**DOI:** 10.3390/nu16234043

**Published:** 2024-11-26

**Authors:** Karin Riemann-Lorenz, Susan Seddiq Zai, Anne Daubmann, Jana Pöttgen, Christoph Heesen

**Affiliations:** 1Institute of Neuroimmunology and Multiple Sclerosis (INIMS), University Medical Center Hamburg-Eppendorf, 20251 Hamburg, Germanyj.poettgen@uke.de (J.P.); heesen@uke.de (C.H.); 2Institute of Medical Biometry and Epidemiology, University Medical Center Hamburg-Eppendorf, 20251 Hamburg, Germany; a.daubmann@uke.de

**Keywords:** multiple sclerosis, food literacy, nutrition knowledge, patient-reported knowledge, outcome measures

## Abstract

Background/Objectives: Persons with MS (pwMSs) are often confronted with contradictory dietary advice, which is not always based on sound scientific evidence. This may lead to poor MS-specific nutrition knowledge (MSNK) and food literacy (MSFL). To date, no studies have assessed MSNK and MSFL among pwMSs. Moreover, no validated tools to measure the effects of educational interventions are available. The aim of this study was to develop and validate MS-specific instruments to measure MSNK and MSFL among pwMSs. Methods: Based on a validated food literacy (FL) screener for the general population and prior research about the information needs of pwMSs, we developed 14 MSFL items and 11 MS-specific nutrition knowledge questions. Cognitive debriefing was conducted with 10 pwMSs and resulted in a 12-item MS food literacy questionnaire (MSFLQ) and an 11-item MS nutrition knowledge questionnaire (MSNKQ). After refinement, both questionnaires were pilot tested in an online survey to explore their comprehensibility. The MSNKQ was analyzed descriptively (mean and percentage of correctly answered questions). For MSFLQ item difficulty, the discriminatory power of the items, internal consistency and convergent/divergent validity were assessed. Results: In total, 148 pwMSs (age: 47.1 years (SD = 12.5); 102 women (69%)) completed the online survey. On average, participants answered 3.51/11 MSNK questions correctly (31.9%). The MSFLQ showed good internal consistency (Cronbach’s alpha = 0.85), item difficulty was good and the discriminatory power of the items was satisfactory. Correlations between the MSFLQ and a general food literacy questionnaire was high (r = 0.626, *p* < 0.001), but only small with the MSNKQ (r = 0.180; *p* = 0.029), underlining the different constructs. Conclusions: MSNK among pwMSs in Germany is low. The MSNKQ and MSFLQ appear to be suitable instruments to assess MSNK and MSFL and might serve as outcome measures for educational interventions.

## 1. Introduction

Multiple sclerosis (MS) is a chronic inflammatory and neurodegenerative disease causing disability in early adulthood [1]. Over the past few decades, the prevalence and incidence of MS have increased worldwide [2,3]. However, the etiology of MS is still only partly known [4], and currently there is no cure for the disease. Research suggests that health behaviors might have an impact on inflammatory disease activity and MS progression [5]. In addition to treatment with disease-modifying therapies (DMTs), adjustment of health behaviors is of growing importance in MS care [6]. Moreover, maintaining a healthy lifestyle appears to be a key factor for successful self-management among persons with MS (pwMSs) [7]. Several studies from different countries have shown the keen interest of pwMSs in dietary modifications [8,9,10,11] and many seek dietary information to manage their condition (e.g., prevent relapses and slow down disease progression) and potentially alleviate symptoms [8,9].

However, systematic reviews have repeatedly shown that there is no convincing RCT-level evidence for specific dietary interventions (e.g., special diets, supplementation of single nutrients) to improve MS outcomes to date [5,10,12,13,14]. A recent systematic review and meta-analysis on the impact of dietary interventions on fatigue and quality of life (QoL) among pwMSs showed reductions in fatigue and improvements in QoL compared to controls for some diets (e.g., Mediterranean, Paleolithic); however, the overall credibility of the evidence was graded as very low due to a high or moderate risk of bias, small sample sizes and the limited number of studies included in the network meta-analysis [15]. In this situation, many researchers in the field conclude that the focus should be more on the effects of foods [14] and complex dietary patterns [16] that are able to influence comorbidities such as obesity, high blood pressure, dyslipidemia and cardiovascular diseases, which in turn can have a negative impact on MS progression [17,18,19]. Recently, this rationale was also adopted by the U.S. National MS Society, which recommended a healthy diet as adjunct care for pwMSs [20].

PwMSs often find it difficult to identify reliable and evidence-based MS-specific nutritional recommendations [21]. They are confronted with a wide variety and often contradictory dietary advice on the Internet, which is often based on anecdotal evidence or conceptual considerations rather than rigorous studies [22]. The lack of evidence increases the risk that pwMSs adopt potentially harmful diets that may cause nutritional deficiencies [13]. Moreover, contradictory information on the Internet or even from health professionals [21] may lead to poor MS-specific nutrition knowledge (MSNK) and low MS-specific food literacy (MSFL), which could make it more difficult for pwMSs to make informed decisions and successfully manage their disease regarding their dietary habits.

Food literacy in general can be regarded as a specific form of health literacy [23]. It comprises the knowledge and skills necessary to find, understand and critically evaluate nutritional information, but also to apply this information when making food choices and preparing meals, thus taking the effects of food choice on personal health and society into account [24,25]. A possible problem with both concepts (i.e., health and food literacy) is that the measurement results are usually based on self-assessment. When answering questions that require self-assessment of knowledge, skills and abilities, individuals will think about different situations and contexts. It is therefore possible that people overestimate or underestimate their food literacy [26]. Krause et al. [26] therefore recommend that the measurement of self-rated food literacy should be complemented by the measurement of nutritional knowledge questions (true/false) and the assessment of food choice.

In response to this situation, several research groups [27,28], including our own [21], have developed and evaluated information and education resources on diet and MS for pwMSs. However, outcomes in these studies did not include MSNK and MSFL, as validated MS-specific questionnaires were not available. The aim of this study was therefore to develop and validate outcome measures that can be used to measure the effects of future educational interventions that aim at increasing MSNK and MSFL.

## 2. Materials and Method

This work is part of the German Research Foundation (DFG)-funded project BEHAVIMS (Health Behavior of persons with Multiple Sclerosis in Germany: Current status and development of supportive strategies for smoking cessation and dietary behavior change). As part of the project, an evidence-based patient information and behavioral change intervention targeting diet and MS will be developed and tested in a multiphase mixed-methods study based on the MRC framework for the development and evaluation of complex interventions [29]. The aim is to increase MS-specific nutrition knowledge, promote MS-specific food literacy and thus empower pwMSs to make informed decisions about their diet.

The project phase described here comprises the development and testing of two instruments to measure MSNK and MSFL among pwMSs. Our research process was guided by recommendations for scale development [30,31]. We developed items based on previous research [21,26], discussed them in our research group of MS experts (i.e., a nutritionist, health scientists, a study nurse, psychologists and neurologists), conducted cognitive interviews with pwMSs and revised items according to the feedback. After that, we collected data from pwMSs in a cross-sectional survey and analyzed psychometric properties (e.g., item analysis), internal consistency, construct validity and scale structure for the MSFL instrument.

### 2.1. Development Process

Fourteen MS-specific food literacy items were developed based on the short food literacy questionnaire (SFLQ) for the general population by Krause et al., 2018 [26]. The SFLQ is a validated questionnaire in the German language measuring functional, interactive and critical dimensions of food literacy. The original questionnaire items were edited, modified and reworded to be MS-specific, i.e., “When I have questions on healthy nutrition, I know where I can find information on this issue” was changed to “When I have questions on healthy nutrition in MS, I know where I can find information on this issue” (for a comparison of the original SFLQ items and the newly developed MS-specific items, see Appendix A). Depending on the question, answers could be given on a four-, five-, six- or seven-point Likert scale that offered the choices “very bad” to “very good”, “disagree strongly” to “agree strongly”, “very difficult” to “very easy”, “very hard” to “very easy”, or “never” to “always”. Some questions also included optional answers like “I don’t do this” or “I am not interested in such information”.

Items on MSNK were generated based on previous research of our group [21], which identified topics of interest and information needs among pwMSs through qualitative and quantitative research methods, collected and assessed the existing evidence on the relationship between nutrition and MS, and tested the feasibility of the evidence-based patient information about diet and MS that we developed [21]. The results showed that MS-specific evidence on the effect of food groups, nutrients and special diets on the MS disease course was of most interest to pwMSs. In contrast, the evidence synthesis showed that based on systematic reviews, there is no convincing RCT-level evidence for specific dietary interventions (e.g., special diets, supplementation of single nutrients) to improve MS outcomes to date [5,10,12,13,14,15]. In line with many researchers in the field and the U.S. National MS Society [20], we concluded and communicated in the evidenced-based patient information [21] that the focus should be more on the effects of healthy dietary patterns [14,16]. These are able to influence the risk of comorbidities such as obesity, high blood pressure, dyslipidemia and cardiovascular diseases, which in turn can have a negative impact on MS progression [17,18,19,20].

Based on this research, eleven knowledge questions were developed with the aim of objectifying whether perceived, self-assessed MSFL matched the actual knowledge (i.e., correctly answered knowledge questions based on scientific evidence). The knowledge questions were in the form of either true or false statements on diet and MS. Participants had to decide between “true”, “false” and “I don’t know” for each statement. “I don’t know” was included as an option to discourage guessing.

### 2.2. Cognitive Debriefing

To assess feasibility and comprehension, the items were qualitatively evaluated via cognitive interviews with pwMSs, using probing techniques, i.e., comprehension probing, category selection probing and general probing [32]. Interviews were conducted in June and July 2023. Participants were pwMSs with a definite MS diagnosis according to the 2017 McDonald criteria [33] recruited via the MS-specific newsletter of the MS Outpatient Clinic of the University Medical Center Hamburg-Eppendorf (UKE).

The interviews were performed by SSZ, audiotaped, transcribed verbatim and analyzed by KRL and SSZ using the software MAXQDA 2022. The assessment included the difficulty of the knowledge items as well as the comprehension and wording of all items. Some items were revised by KRL and SSZ according to the feedback received; all changes were discussed in our multi-disciplinary research group of MS experts. The improved items were then again assessed for the difficulty of the knowledge items and the wording and comprehension of the MSFL items by three pwMSs recruited in the MS Outpatient Clinic of the UKE.

### 2.3. Web-Based Survey

#### 2.3.1. Procedure

A web-based survey using the data-protected survey platform LimeSurvey provided by the University of Hamburg was conducted to evaluate the psychometric properties of the two questionnaires. For the survey, pwMSs were recruited via the newsletter of the MS Outpatient Clinic of the UKE. The link to the survey was sent out in December 2023; the survey was closed in January 2024. PwMSs were able to participate if they were aged ≥18 years and had an MS diagnosis.

Written informed consent was obtained online from all participants before starting the assessment.

#### 2.3.2. Measures

We collected demographic data (i.e., age, sex, height, weight, smoking status) and disease-specific data (i.e., time since diagnosis, MS subtype and data on medication).

To evaluate the convergent validity of the MS food literacy questionnaire (MSFLQ), we administered the following instruments:The German version of the validated Spanish short Diet Quality Screener (sDQS) [34,35,36]. The sDQS is a measure to estimate the overall quality of the dietary pattern for use in time-limited settings like primary care. The study participants are asked to report their average consumption of 17 food groups over the last four weeks. To calculate the German version of the sDQS score with a possible range from 17 to 51, all food item scores were summed up. A higher score indicates a more favorable dietary pattern and greater adherence to the dietary recommendations of the German Nutrition Society for the general population [37].The Health Literacy Survey (HLS-GER) subscales “coping with illness” and “health promotion” [38]. The HLS-GER measures general health literacy in the German population. Participants report on their self-rated ability to find, understand, evaluate and apply health-related information.The German Food Literacy Score (FLS) is used to assess general food literacy. The FLS was developed on the basis of the Self-Reported Food Literacy Questionnaire by Poelman et al., 2018 [39]. The FLS covers eight dimensions of food literacy that relate to the necessary knowledge, skills and behaviors that are needed to plan and organize meals and to select and prepare foods to meet the individual’s physiological and psychological needs. The German FLS consists of 29 items and has been used in a representative survey in Germany in 2020 [40].

To assess divergent validity, theBrief Illness Perception Questionnaire (IPQ) [41] was used.

The IPQ is a validated questionnaire used to assess illness perceptions, with a focus on cognitive illness representation and emotional illness representation.

Additionally, all participants were asked for feedback concerning duration, possible emotional distress elicited by the questions and the difficulty of the items after completing the MSFLQ and the MS nutrition knowledge questionnaire (MSNKQ).

### 2.4. Statistical Analyses

Descriptive analyses of demographic and clinical data were performed. Quantitative variables are described by the mean and standard deviation (SD) and/or the median and range. Categorial variables are expressed using absolute and relative frequencies.

First, to determine the psychometric properties of the MSFLQ, item difficulty [42] and internal consistency (Cronbach’s alpha) were assessed. An item difficulty between >0.1 and <0.9 is considered satisfactory [42]; for Cronbach’s alpha, a value > 0.7 is needed for satisfactory internal consistency [43]. The discriminatory power of the items was assessed; values below 0.30 are considered low, while values above 0.50 are considered high [44].

To find the underlying constructs, two confirmatory factor analyses (CFAs) were carried out. The dimensionality of the questionnaire was first calculated to confirm whether the scale had one dimension (as seen in the original SFLQ by Krause et al.) [26] or three dimensions, as seen in the theoretical model by Nutbeam et al., on which the original SFLQ by Krause et al. had been developed [23,45]. Afterwards an exploratory factor analysis (EFA) was performed to further explore dimensionality. A principal component analysis was selected through orthogonal varimax rotation to achieve the simplest reasonable test structure. To determine the number of factors, an Eigenvalue > 1 [46] and an explained variance of over 10% was used [47].

For the validity analysis (convergent/divergent), correlations were performed using Pearson’s correlation coefficient to investigate associations between the assessed measures (i.e., the sDQS, the FLS, the HLS and the IPQ) and the MSFLQ. Based on guidelines [48], associations of 0.1, 0.3 and 0.5 were considered small, moderate and large, respectively.

Second, the MSNKQ was evaluated descriptively. The balance between difficult and easy questions was analyzed by examining the percentage of correct answers, and the average number (range) of correct answers was determined.

All analyses were performed using the Statistical Package for the Social Sciences (IBM SPSS Statistics, SPSS Version 27, Chicago, IL, USA).

### 2.5. Ethical Statement

This study was approved by the Ethics Committee of the Hamburg Chamber of Physicians (No. 2022-100779-BO-ff) and is in accordance with the Declaration of Helsinki.

## 3. Results

### 3.1. Development of Questionnaires

#### 3.1.1. Development of MSFLQ

The first version of the MSFLQ consisted of 14 questions on food literacy. The questions focused on the participant’s perceived ability in finding, understanding, selecting, applying, exchanging and critically evaluating information on diet and MS. Additionally, two questions evaluated the practical implementation of this MS-specific nutritional knowledge in daily life. The total score was determined by adding up the points from the individual items, with a higher score indicating higher perceived food literacy.

#### 3.1.2. Development of MSNKQ

The MSNKQ had 11 questions. The questions focused on MS-specific dietary recommendations regarding food intake, supplement use, vitamin D and the influence of comorbidities on MS progression. Each question answered correctly was awarded one point; questions answered incorrectly or with “don’t know” were awarded zero points. The total score was calculated by adding up the number of questions answered correctly, which resulted in a possible range from 0 to 11.

### 3.2. Cognitive Interviews

The first version of the MSFL items and the MSNK items was administered to seven pwMSs (n = 5 female) with an age range from 32 to 68 years. Using probing techniques, it took participants 30 to 60 min to complete the debriefing.

The feedback on the MSFLQ was generally positive; the questions were rated as understandable and acceptable. Minor changes in wording were made. In question two, we changed “verbal recommendations” to “verbal information”; in question six, we changed “important information” to “relevant information”. For question seven, the meaning of “individual foods” was unclear for one participant, so we added the phrase “for example cow’s milk, pork or chia seeds” to the item for clarification. For question eight, we specified food groups by adding the phrase “for example vegetables, fish or meat” and changed the term “health-promoting diet” to “healthy diet”. In question 11, we changed “I find it easy to maintain a predominantly plant-based diet” to “it’s easy for me to eat significantly more plant-based foods than animal-based foods”. Two questions on discussing dietary recommendations for MS with others were excluded, because question 12 was also related to this topic and perceived as the best of those three questions by the participants. These changes resulted in a 12-item MSFLQ with a possible score range from 2 to 52. The original items for cognitive debriefing as well as the final items of the MSFLQ can be found in Appendix A.

For the knowledge questions, we made several changes to the wording. In item one, we changed “eat a plant-based diet” to “follow a predominantly plant-based diet” and “to avoid dairy products completely” to “avoid cow’s milk and dairy products”. Item three was rephrased from “People with MS should avoid foods that contain milk or gluten” to “The consumption of foods containing gluten (e.g., bread or pasta) should be avoided if you have MS”. For item four, we added “e.g., cow’s milk, pork or chia seeds” for clarification. The wording for item five was changed from “[it] is recommended” to “people with MS should” and the term “nutritional supplements” was removed. For item seven, we slightly changed the wording. Two items on the association of vitamin D and MS were removed because participants preferred item six which also described the possible effect of vitamin D. Two items were added to the knowledge questionnaire: a question on omega-3 supplementation, after we saw gaps in knowledge in the cognitive debriefing, when a participant commented on omega-3 supplementation, and a question on obesity. The final items of the MSNKQ can be found in Appendix A.

After editing the questionnaires, we performed another cognitive debriefing with three additional participants. We found no need for further modifications afterwards.

### 3.3. Survey Results

For the web-based survey, a total of 187 participants gave informed consent. A total of 148 pwMSs finished the survey and were included in the analysis. The sociodemographic and disease-related characteristics of the participants are displayed in Table 1.

#### 3.3.1. MSFLQ

Participants scored on average 34.89 (SD = 7.8; median: 35.83; range: 8–52) points. The scale showed good internal consistency (Cronbach´s alpha: 0.85; Cronbach’s alpha if one item left out: range 0.83–0.85) and satisfactory discriminatory power of the items (≥0.30) (Table 2). Item difficulty ranged between 0.49 and 0.80.

The overall feedback on the MSFLQ was generally good. Filling out the questionnaire was not emotionally stressful or too time-consuming for most participants. Items were not difficult to understand, nor did they irritate participants. The majority acknowledged that filling out the questionnaire made them curious to look (more) into the topic. For more details, please see Appendix A.

Neither a three-factor scale structure nor a one-factor scale structure was confirmed, so we decided to further explore dimensionality via EFA. Items were loaded on two factors (Table 3), which were Eigenvalues > 1 and explained variance > 10%. The two factors explained 54% of the total variance. All items except for item 10 and item 11 loaded on factor 1.

The results of the convergent/divergent validity assessment are shown in Table 4. The MSFLQ correlated highly with both the FLS and the HLS-GER. Correlation with the sDQS was moderate to high. The correlation between the MSFLQ and the MSNKQ was small. Divergent validity was assessed by correlating the MSFLQ with the IPQ, which showed a low-to-moderate correlation (see Table 4).

#### 3.3.2. MSNKQ

Survey participants answered on average 3.51 of 11 (31.9%) questions correctly (SD 2.4; range 0–11). The range of correct answers per item was from 6.8% to 77%. The highest rate of correct answers was seen for item 11 (A diet that reduces the risk for cardiovascular disease can also have a positive impact on the course of MS), with 77% correct answers, followed by item 8 (Being obese has a detrimental impact on the course of MS.), with 70% correct answers (Table 5).

The two most challenging questions (highest rate of wrong/don’t know) were item 10 (Taking Omega-3 fatty acid capsules has a beneficial effect on the course of MS) and item 4 (Individual foods (e.g., cow’s milk, pork or chia seeds) can have an influence on the course of MS).

The overall feedback on the MSNKQ was good. Filling out the questionnaire was neither irritating nor emotionally stressful, and the items were not difficult to understand. The majority acknowledged that filling out the questionnaire made them curious to look (more) into the topic, and about half of the participants stated that many questions were difficult to answer, because they were lacking the necessary knowledge. For more details, please see Appendix A.

## 4. Discussion

This study described the development and validation of an MS-specific food literacy questionnaire. In line with recommendations [26], we also developed an MS nutrition knowledge questionnaire based on previous research of our group [21] to complement self-rated MSFL with data on actual MSNK. The results provide insights into the level of self-perceived MSFL and the actual MSNK of pwMSs recruited from the MS Outpatient Clinic of a university medical center in Germany.

The 12 items of the MSFLQ cover aspects of functional, interactive and critical FL. It is noticeable that the participants rated their food literacy skills particularly low on two items of the MSFLQ which represent interactive and critical FL. Item 12 asks how easy it is to talk to others about nutrition for MS (interactive FL). Why pwMSs in this study perceive this as rather difficult is hard to say and requires further research. Item seven asks how easy it is to assess the truthfulness of claims about connections between food and the course of MS (critical food literacy). The rather low scores on this item are in line with results from previous studies which demonstrated that pwMSs find it difficult to assess the accuracy and trustworthiness of such information [21,49,50]. Also, the rather low scores on item one (i.e., the ability to find reliable information) aligns well with these results. Overall, a mean value of ~35 on a scale with a maximum of 52 points (perfect MSFL) demonstrates that the group of pwMSs in this study does not rate their own MSFL as unproblematic, but indicates potential for improvement possibly by providing reliable and trustworthy information.

The EFA identified a two-dimensional structure of the MSFLQ instead of one dimension, as previously shown by Krause et al. [26], or three dimensions, as was conceptualized by Nutbeam’s model of functional, interactive and critical health literacy [23,45]. The two items loading on the second factor were item 10 (“It’s easy for me to prepare a vegetarian main meal”) and item 11 (“In everyday life, it’s easy for me to eat significantly more plant-based foods than animal-based foods”). Those items would originally be part of functional literacy in Nutbeam’s model, which could be summarized as finding information, understanding that information, knowing it and being able to apply that knowledge when choosing foods or preparing meals [51]. We found in our study that actually applying knowledge might be a different ability than finding, understanding and knowing information, as it requires other skills (i.e., being able to cook, planning meals). All other items loaded on one factor, similarly to the SFLQ [26].

Internal consistency was good. Therefore, we were able to calculate an MSFLQ sum score and investigate the construct validity of the instrument. Overall, the construct validity of the MSFLQ was adequate, because we found large correlations with general food literacy, general health literacy and the quality of the dietary pattern. However, the correlation of the MSFLQ with the MSNKQ was small to moderate. Normally, a high correlation between food literacy and nutrition knowledge would be expected, because nutrition knowledge can be considered part of FL [24]. As expected, we only found low-to-moderate correlations of the MSFLQ with the IPQ.

On average, the study participants answered only 32% of the MSNK questions correctly. If school grades were used as a benchmark for evaluating this result, it would indicate insufficient knowledge or failure. However, the range of correctly answered questions was between 0 and 11, which points to great differences in the level of knowledge. We identified tremendous gaps in knowledge in most questions that addressed the possible influence of single nutrients or foods on MS disease course or specific recommendations on food choice or supplement use. Questions around the influence of comorbidities on MS were more often answered correctly. These results support our hypothesis that contradictory dietary advice on the Internet, which is often based on anecdotal evidence, conceptual considerations or commercial interests rather than rigorous studies, may cause poor MSNK. In particular, pwMSs who search extensively for nutrition information on the Internet may therefore rate their own food literacy as high, while considering non-evidence-based information to be correct, and consequently may have achieved low scores in the MSNKQ. This might be one explanation for the weak correlation between the MSNKQ scores and the MSFLQ scores in our study.

The dropout rate was low (21%) and most participants did not perceive filling in the questionnaires as stressful or too time-consuming. Hence, both questionnaires proved to be practical and were well accepted by the participants of the validation study.

### Strengths and Limitations

The study results must be interpreted with caution due to some limitations. The sample might only be representative for the general MS population in Germany to a limited extent, as most participants were well educated, had contact to the outpatient clinic of a university medical center, and on average had only mild-to-moderate disability. Moreover, only pwMSs who were interested in the survey took part. Also, the study results might not be generalizable to pwMSs in other countries, and the applicability of the developed questionnaires could be limited, especially if food habits and information resources available for this patient group differ substantially from those in Germany. Furthermore, we did not examine the test–retest reliability and responsiveness of the questionnaires, so further studies to assess these aspects are needed.

A strength of this study is that it was guided by existing recommendations for scale development. The MSFLQ was based on an existing and validated general food literacy questionnaire, which increases the credibility of our results. Also, following the recommendation of Krause et al. [26], we not only measured self-assessed MSFL but also—with the MSNKQ—constructed a knowledge questionnaire to objectify self-assessed MSFL.

## 5. Conclusions

To the best of our knowledge, the MSFLQ and the MSNKQ are the first validated questionnaires that allow us to empirically assess and compare self-reported MSFL and actual MSNK among pwMSs. It may be used to evaluate the effects of educational interventions in this population and thereby increase our understanding of MSFL and MSNK among pwMSs. The low MSNK scores in this study highlight the need for providing this patient group with evidence-based information on nutrition and MS. This will eventually increase MSNK, reduce uncertainty and support pwMSs to make informed decisions about their diet.

## Figures and Tables

**Table 1 nutrients-16-04043-t001:** Sociodemographic and disease-related characteristics.

	pwMSs (N = 148)
**Age (years)**	
Mean (SD)	47.14 (12.5)
[Min–Max]	[19–78]
**Sex**	
Male	46 (31.1%)
Female	102 (68.9%)
**Education**	
<12 years	41 (27.7%)
≥12 years	107 (72.3%)
**Employment**	
Full-time	54 (36.5%)
Part-time	38 (25.7%)
In training/school	4 (2.7%)
Unemployed	5 (3.4%)
Pension (disability, old age)	41 (27.7%)
Other (i.e., sick leave, self-employed, homemaker)	6 (4.1%)
**Disease duration (years)**	
Mean (SD)	9.70 (8.9)
[Min–Max]	[0–35]
**MS type**	
RRMS	83 (56.1%)
SPMS	27 (18.2%)
PPMS	21 (14.2%)
Unclear	17 (11.5%)
**DMT use**	
No	55 (37.2%)
Yes	87 (58.8%)
In the decision-making process	6 (4.1%)
**PDDS**	
Mean (SD) [Min–Max]	2.27 (1.9) [0–7]
No disability	34 (23.0%)
Mild disability	24 (16.2%)
Moderate disability	26 (17.6%)
Gait disability	28 (18.9%)
Early cane	17 (11.5%)
Late cane	9 (6.1%)
Bilateral support	7 (4.7%)
Wheelchair	3 (2.0%)
Bedridden	0 (0%)
**Smoking**	
Never smoked	73 (49.3%)
Still smoking	18 (12.2%)
Smoked in the past	57 (38.5%)
**BMI**	
Mean (SD) [Min–Max]	24.09 (4.2) [18–47]
Underweight (BMI: <18.5)	6 (4.1%)
Normal weight (BMI: 18.5–24.9)	89 (60.1%)
Overweight (BMI: 25–29.9)	44 (29.7%)
Obesity (BMI: >30)	9 (6.1%)

RRMS = Relapsing Remitting Multiple Sclerosis; SPMS = Secondary Progressive Multiple Sclerosis; PPMS = Primary Progressive Multiple Sclerosis; DMT = Disease-Modifying Therapy; PDDS = Patient-Determined Disease Steps Scale; BMI = Body Mass Index.

**Table 2 nutrients-16-04043-t002:** Descriptive results and psychometric properties of MSFLQ.

Item		Min–Max	Mean (SD)	Item Difficulty	Discriminatory Power	Cronbach’s Alpha, If One Item Left Out
MSFLQ1	If I have queries about nutrition for MS, I know where to find reliable information.	Does not apply at all = 1 to Applies completely = 4; I have no experience with this = 0	2.49 (1.1)	0.62	0.54	0.836
MSFLQ2	In general, how well do you understand the following information about nutrition for MS? (What is meant is the comprehensibility and not the quality of the information.) Please tick one answer for each line.	Very bad = 1 to Very good = 5; I don’t use it = 0	3.56 (1.3)		-	-
	2-1 Information in brochures or books			0.53	0.55	0.836
	2-2 Information on websites or in podcasts			0.49	0.52	0.838
	2-3 Verbal information from healthcare professionals			0.52	0.40	0.847
MSFLQ3	How familiar are you with the recommendations for healthy eating for people with MS?	Very bad = 1 to Very good = 5; I don’t know them = 0	3.22 (1.2)	0.64	0.67	0.827
MSFLQ4	How well do you manage to select the information that is relevant to you about healthy eating for MS?	Very bad = 1 to Very good = 5; I have no interest in such information = 0	3.39 (1.0)	0.68	0.68	0.829
MSFLQ5	How easy is it for you to judge whether information about nutrition for MS is trustworthy?	Very difficult = 1 to Very easy = 4; I have no interest in such information = 0	2.62 (0.8)	0.66	0.55	0.838
MSFLQ6	How well do you manage to distinguish relevant information about nutrition for MS from not so relevant information?	Very bad = 1 to Very good = 5; I have no interest in such information = 0	3.36 (1.1)	0.67	0.65	0.830
MSFLQ7	On the Internet, individual foods (e.g., cow’s milk, pork or chia seeds) are often associated with an improvement or a deterioration in the course of MS. How easy is it for you to judge to what extent the relationships presented are true or not?	Very difficult = 1 to Very easy = 4; I don’t do that = 0	2.32 (1.0)	0.58	0.49	0.839
MSFLQ8	How easy is it for you to assess the role that different food groups (e.g., vegetables, fish or meat) play in a healthy diet for MS?	Very difficult = 1 to Very easy = 4	2.93 (0.8)	0.64	0.65	0.834
MSFLQ9	How easy is it for you to assess what impact your eating habits could have on your MS in the long term?	Very difficult = 1 to Very easy = 4	2.72 (0.8)	0.57	0.42	0.843
MSFLQ10	It’s easy for me to prepare a vegetarian main meal.	Does not apply at all = 1 to Applies completely = 4; I don’t do that = 0	3.20 (1.1)	0.80	0.30	0.850
MSFLQ11	In everyday life, it’s easy for me to eat significantly more plant-based foods than animal-based foods.	Does not apply at all = 1 to Applies completely = 4; I don’t do that = 0	3.04 (1.0)	0.76	0.30	0.849
MSFLQ12	How easy is it for you to exchange ideas with others about nutrition for MS?	Very difficult = 1 to Very easy = 4; I don’t do that = 0	2.01 (1.5)	0.50	0.42	0.846

MSFLQ = MS-specific food literacy questionnaire.

**Table 3 nutrients-16-04043-t003:** Exploratory factor analysis (EFA) of MSFLQ (N = 148) (principal component analysis with orthogonal varimax rotation).

Item	h^2^	Factor 1	Factor 2
Item 8	0.694	0.826	
Item 4	0.668	0.802	
Item 6	0.645	0.783	
Item 3	0.617	0.776	
Item 5	0.525	0.693	
Item 1	0.452	0.679	
Item 7	0.386	0.663	
Item 9	0.466	0.594	
Item 12	0.271	0.429	
Item 2	0.185	0.405	
Item 11	0.820		0.903
Item 10	0.783		0.878

h^2^ = communality. Factor loadings < 0.4 are not displayed.

**Table 4 nutrients-16-04043-t004:** Results of MSFLQ—convergent and divergent validity (N = 148).

Questionnaires	MSFLQ
FLS	0.626 **
HLS-GER	0.541 **
sDQS	0.461 **
MSNKQ	0.180 *
IPQ	−0.294 **

FLS = Food Literacy Score; HLS-GER = Health Literacy Survey (Germany); sDQS = short Diet Quality Screener; MSNKQ = MS nutrition knowledge questionnaire; IPQ = Illness Perception Questionnaire. * The correlation is significant at the level of 0.05 (2-sided). ** The correlation is significant at the level of 0.01 (2-sided).

**Table 5 nutrients-16-04043-t005:** Absolute and relative frequencies of correct and wrong answers to MSNKQ items.

Question	Correct n (%)	Wrong n (%)	I Don’t Know n (%)	Wrong/Don’t Know n (%)
1	33 (22.3%)	92 (62.2%)	23 (15.5%)	115 (77.7%)
2	43 (29.1%)	73 (49.3%)	32 (21.6%)	105 (70.9%)
3	65 (43.9%)	41 (27.7%)	42 (28.4%)	83 (56.1%)
4	19 (12.8%)	86 (58.1%)	43 (29.1%)	129 (87.2%)
5	21 (14.2%)	80 (54.1%)	47 (31.8%)	127 (85.8%)
6	28 (18.9%)	88 (59.5%)	32 (21.6%)	120 (81.1%)
7	27 (18.2%)	16 (10.8%)	105 (70.9%)	121 (81.8%)
8	104 (70.3%)	2 (1.4%)	42 (28.4%)	44 (29.7%)
9	56 (37.8%)	60 (40.5%)	32 (21.6%)	92 (62.2%)
10	10 (6.8%)	86 (58.1%)	52 (35.1%)	138 (93.2%)
11	114 (77.0%)	3 (2.0%)	31 (20.9%)	34 (23.0%)

## Data Availability

The raw data supporting the conclusions of this article will be made available by the authors on request.

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
