# Peer review of "Nutrition Knowledge and Food Literacy Among Persons with Multiple Sclerosis—Development and Validation of Patient-Reported Outcome Measures"

_nutrients, 2024, doi:10.3390/nu16234043_

Round 1
Reviewer 1 Report
Comments and Suggestions for Authors
very good publication, I only have a few comments:
-line 24: missing unit for age result. It should be e.g. years old.
-For the questionnaires used I miss a cut-off point/some point range and interpretation of the result. Especially since later in the discussion you write "On average, only 32% of the knowledge questions were answered correctly indicating a low level of MSNK." On what basis do you consider the result to be low?
-In addition to limitations it is also worth writing about the strengths of the articles.
-Line 262: missing period
Author Response
Dear Ms Wang, dear reviewer,
thank you very much for the opportunity to revise our manuscript.
Thank you also for your appreciation and your valuable comments. Below you will find our point by point reply to all suggestions and issues.
Reviewer 1:
very good publication, I only have a few comments:
Comment 1.1: line 24: missing unit for age result. It should be e.g. years old.
Reply: We added the word “years” in line 24.
Comment 1.2: For the questionnaires used I miss a cut-off point/some point range and interpretation of the result. Especially since later in the discussion you write "On average, only 32% of the knowledge questions were answered correctly indicating a low level of MSNK." On what basis do you consider the result to be low?
Reply: Thank you for this comment. Defining cutoffs or clinically meaningful change is not trivial. As this is the first knowledge questionnaire on diet in MS and to our knowledge no data exist about dietary knowledge among pwMS, we cannot provide cutoffs in relation to other tools. In general less than 50% correct replies would be considered failure in a knowledge test and therefore the results are definitely poor. In the past, we have developed several knowledge questionnaires in MS (RIKNO - DOI:10.1371/journal.pone.0138364 and DOI: 10.1371/journal.pone.0208004; MRI knowledge - Schiffmann I et al., The MRI-risk knowledge questionnaire 2.0 (MRI-RIKNO 2.0) - assessment of MRI specific knowledge in people with multiple sclerosis and its association with emotions towards MRI. Ectrims Online Library. (2019) 279618 and DOI:10.3389/fneur.2022.856240; and MOMS -Peper et al. - DOI:10.1016/j.msard.2023.104789, where we encountered similar problems.
To respond to your comment, we now tried to provide a clearer interpretation of the results from our perspective in the discussion.
Line 395+ referring to the MSNK results is now reading:
„On average, the study participants answered only 32% of the MSNK questions correctly. If school grades were used as a benchmark for evaluating this result, it would indicate insufficient knowledge or failure. However, the range of correctly answered questions was between 0 and 11, which points to great differences in the level of knowledge.“
Line 367+ referring to the MSFL results is now reading:
„Overall, a mean value of ~35 on a scale with a maximum of 52 points (perfect MSFL) demonstrates that the group of pwMS in this study does not rate their own MSFL as unproblematic, but indicates potential for improvement possibly by providing reliable and trustworthy information.“
Comment 1.3. In addition to limitations it is also worth writing about the strengths of the articles.
Reply: We agree with your comment and included the following sentences in the limitations section, which we renamed Strengths and Limitations (line 429+).
A strength of this study is that it was guided by existing recommendations for scale development. The MSFLQ was based on an existing and validated general Food Literacy Questionnaire, which increases the credibility of our results. Also, following the recommendation of Krause et al. [26], we not only measured self-assessed MSFL but also – with the MSNKQ – constructed a knowledge questionnaire to objectify self-assessed MSFL.
Comment 1.4: Line 262: missing period
Reply: We added a period at the end of the sentence, now in line 280.
Reviewer 2 Report
Comments and Suggestions for Authors
There is accumulating epidemiological and experimental evidence that diet can influence the prognosis of patients with multiple sclerosis, and previous researches suggested (to some extent) what is considered appropriate diet for them. I consider this research paper is unique and important, because it asks patients about their knowledge of diet and nutrition in relation to MS.
The questionnaire development process, content and results are generally carefully described and easy to understand; the reviewer did not fully understand the section on the MSNKQ. A more comprehensive description is needed.
3.1.2 Development of the MSNKQ
What evidence was used to develop these 11 questions?
What are the correct answers to each question? What is the evidence?
Ref.12 is cited but please explain in more detail on these questions.
Food habits differ from country to country and it is difficult to generalise globally. (This reviewer is not from Western countries. Maybe that's why I feel so more.) In addition, as there are individual differences, strong recommendations such as ‘should follow’ do not seem to fit.
Reference:
https://www.nationalmssociety.org/managing-ms/living-with-ms/diet-exercise-and-healthy-behaviors/diet-nutrition
Author Response
Dear Ms Wang, dear reviewer,
thank you very much for the opportunity to revise our manuscript.
Thank you also for your appreciation and your valuable comments. Below you will find our point by point reply to all suggestions and issues.
Comment 2.1: There is accumulating epidemiological and experimental evidence that diet can influence the prognosis of patients with multiple sclerosis, and previous researches suggested (to some extent) what is considered appropriate diet for them. I consider this research paper is unique and important, because it asks patients about their knowledge of diet and nutrition in relation to MS.
The questionnaire development process, content and results are generally carefully described and easy to understand; the reviewer did not fully understand the section on the MSNKQ. A more comprehensive description is needed.
3.1.2 Development of the MSNKQ
Reply: Thank you for your comment. We now added more detail to the development process of the MSNKQ. Line 122+ is now reading:
„Items on MSNK were generated based on previous research of our group [21], which identified topics of interest and information needs among pwMS through qualitative and quantitative research methods, collected and assessed the existing evidence on the relationship between nutrition and MS and tested the feasibility of the evidence-based patient information about diet and MS we developed [21]. Results showed that MS-specific evidence on the effect of food groups, nutrients and special diets on the MS disease course was of most interest to pwMS. In contrast, the evidence synthesis showed that based on systematic reviews there is no convincing RCT level evidence for specific dietary interventions (e.g. special diets, supplementation of single nutrients) to improve MS outcomes to date [5, 10, 12-15]. In line with many researchers in the field and the U.S. National MS Society [20] we concluded and communicated in the evidenced-based patient information [21] that the focus should be more on the effects of healthy dietary patterns [14, 16]. These are able to influence the risk for comorbidities such as obesity, high blood pressure, dyslipidemia and cardiovascular diseases, which in turn can have a negative impact on MS progression [17-20].
Based on this research, eleven knowledge questions were developed with the aim to objectify whether perceived, self-assessed MSFL matched with actual knowledge (i.e., correctly answered knowledge questions based on scientific evidence).“
Comment 2.2: What evidence was used to develop these 11 questions?
Reply: Please see our answer to your first comment (2.1)
Comment 2.3: What are the correct answers to each question? What is the evidence? Ref.12 is cited but please explain in more detail on these questions.
Reply: Thank you for your questions. We now added the correct answers to each knowledge question in the Supplement Table S3. For the evidence base we kindly ask you to look at our response to your first comment (2.1) and the included references.
Comment 2.4: Food habits differ from country to country and it is difficult to generalise globally. (This reviewer is not from Western countries. Maybe that's why I feel so more.) In addition, as there are individual differences, strong recommendations such as ‘should follow’ do not seem to fit.
Reply: Thank you for your comment. We agree that food habits differ in different parts of the world and that there might be individual differences in food intake, nutrient supply and response to individual foods (e.g. if you have a food allergy). However, evidence-based nutrition information attempts to provide generalisable nutritional recommendations for a specific patient group based on the current results of scientific studies. If, for example, the evidence synthesis based on systematic reviews of randomized studies shows that taking nutritional supplements or high-dose vitamin D does not provide any benefit in terms of relevant MS outcomes, we consider the statements we have made to be justified. Of course, this does not rule out the possibility that individual patients may have nutrient deficiencies due to their lifestyle or eating habits. However, finding this out and correcting the deficiencies remains a therapeutic task that can only be solved on an individual basis.
Referring to different food habits from country to country, we now made clearer that the development of the MSNKQ items was based on research with pwMS in Germany (please see answer to your comment 2.1) reflecting their questions and (mis-)conceptions of advisable MS diets. We also added a sentence to the limitations section to acknowledge the limited transferability to countries with different food habits and information resources for pwMS. Line 422+ is now reading:
„Also, the study results might not be generalisable to pwMS in other countries and the applicability of the developed questionnaires could be limited, especially if food habits and information resources available for this patient group differ substantially from those in Germany.“
Reviewer 3 Report
Comments and Suggestions for Authors
Thank you for the opportunity to review this paper. The paper is generally well-written and structured. The authors should consider the following:
The main question addressed by the research is clearly written in the abstract.
Compared with other published material, author’s justification is poor, Why this study and why now? The authors should better explain what new the study/ paper brings to scholarship.
The methodology is not well-justified in the sense that it has to be based on published guidelines. This is lacking at the moment. The authors should clearly present and justify which guidelines for validating questionnaires they used.
The authors claim that there are no validated questionnaires measuring nutrition knowledge among patients with MS. However, there are published studies of nutrition among MS patients. What these questionnaires are about?
The conclusions consistent with the evidence and arguments presented and address the main question posed.
The references are appropriate.
There is no any additional comments on the tables and figures.
Author Response
Dear Ms Wang, dear reviewer,
thank you very much for the opportunity to revise our manuscript.
Thank you also for your appreciation and your valuable comments. Below you will find our point by point reply to all suggestions and issues.
Thank you for the opportunity to review this paper. The paper is generally well-written and structured. The authors should consider the following: The main question addressed by the research is clearly written in the abstract.
Comment 3.1: Compared with other published material, author’s justification is poor, Why this study and why now? The authors should better explain what new the study/ paper brings to scholarship.
Reply: Thank you for your critical comment. We now made clearer why the development of MS-specific food literacy and nutrition knowledge questionnaires is of importance and timely for pwMS and researchers, who develop nutrition information and education resources for pwMS. Line 70+ is now reading:
„Moreover, contradictory information on the Internet or even from health professionals [21] may lead to poor MS-specific nutrition knowledge (MSNK) and low MS-specific food literacy (MSFL), which could make it more difficult for pwMS to make informed decisions and successfully manage their disease regarding their dietary habits.“
Line 83+ is now reading:
„In response to this situation, several research groups [27, 28], including our own [21], have developed and evaluated information and education resources on diet and MS for pwMS. However, outcomes in these studies did not include MSNK and MSFL as validated MS-specific questionnaires have not been available. The aim of this study was therefore to develop and validate outcome measures that can be used to measure the effects of future educational interventions that aim at increasing MSNK and MSFL.“
Comment 3.2: The methodology is not well-justified in the sense that it has to be based on published guidelines. This is lacking at the moment. The authors should clearly present and justify which guidelines for validating questionnaires they used.
Reply:
Thank you for comment. We agree and have included more detail on the recommendations we followed, line 100+ now reading:
„Our research process was guided by recommendations for scale development [30, 31]. We developed items based on previous research [21, 26], discussed them in our research group of MS experts (i.e., a nutritionist, health scientists, a study nurse, psychologists and neurologists), conducted cognitive interviews with pwMS and revised items according to the feedback. After that we collected data from pwMS in a cross-sectional survey and analysed psychometric properties (e.g. item analysis), internal consistency, construct validity and scale structure for the MSFL instrument.
Comment 3.3. The authors claim that there are no validated questionnaires measuring nutrition knowledge among patients with MS. However, there are published studies of nutrition among MS patients. What these questionnaires are about?
Reply: Thank you for your question. We agree that there are published studies of nutrition among persons with MS, but these usually assess dietary habits usings food records (e.g., DOI: 10.1016/j.msard.2024.105467) or dietary screeners (e.g., DOI: 10.1111/ene.15066 ; doi: 10.3389/fneur.2023.1172419) for the assessment of the dietary pattern or nutrient intake. Moreover, there are studies that assess the experiences with and needs for information around diet and MS (e.g. Russell et al, doi: 10.1111/hex.13226; Riemann-Lorenz et al., DOI: 10.1080/09638288.2024.2388259; Russell et al. DOI: 10.1016/j.msard.2024.105816) To the best of our knowledge, we are not aware of questionnaires assessing the actual MS-specific nutrition knowledge among pwMS.
- The conclusions consistent with the evidence and arguments presented and address the main question posed.
Reply: Thank you.
- The references are appropriate.
Reply: Thank you.
- There is no any additional comments on the tables and figures.
Reply: Thank you.
Round 2
Reviewer 2 Report
Comments and Suggestions for Authors
significantly improved
Reviewer 3 Report
Comments and Suggestions for Authors
I think my comments have been addressed